# Absolute marine gravimetry with matter-wave interferometry

Y. Bidel[1], N. Zahzam[1], C. Blanchard[1], A. Bonnin[1], M. Cadoret[1,2], A. Bresson[1], D. Rouxel[3] &
M.F. Lequentrec-Lalancette[3]

Measuring gravity from an aircraft or a ship is essential in geodesy, geophysics, mineral and hydrocarbon exploration, and navigation. Today, only relative sensors are available for onboard gravimetry. This is a major drawback because of the calibration and drift estimation procedures which lead to important operational constraints. Atom interferometry is a promising technology to obtain onboard absolute gravimeter. But, despite high performances obtained in static condition, no precise measurements were reported in dynamic. Here, we present absolute gravity measurements from a ship with a sensor based on atom interferometry. Despite rough sea conditions, we obtained precision below $10^{-5}$ m s$^{-2}$. The atom gravimeter was also compared with a commercial spring gravimeter and showed better performances. This demonstration opens the way to the next generation of inertial sensors (accelerometer, gyroscope) based on atom interferometry which should provide high-precision absolute measurements from a moving platform.

[1] ONERA – The French Aerospace Lab, F-91123 Palaiseau, Cedex, France. [2] Laboratoire Commun de Métrologie, CNAM, 61 Rue du Landy, 93210 La Plaine Saint-Denis, France. [3] Shom – French hydrographic and oceanographic office, CS 92803, 29228 Brest, France. Correspondence and requests for materials should be addressed to Y.B. (email: yannick.bidel@onera.fr)

The precise knowledge of Earth's gravity field is of major importance in several domains. In geodesy, it is essential for describing the continental and sea surface topography, since the geoid, an equipotential surface of gravity, is used as a height reference. In geophysics, gravity measurements provide information on the underground mass distribution and its variations. They allow thus mapping tectonic structure[1], studying volcano[2] and earthquake[3], monitoring ice melting[4], measuring water storage variation[5], or exploring oil, gas, and mineral[6]. The knowledge of gravity field is also essential in inertial navigation as navigation algorithms need a precise gravity model[7].

The Earth's gravity field can be measured from space by using for example satellite to satellite tracking methods[8,9] or satellite-borne gravity gradiometer[10,11]. Nevertheless, these methods lead to gravity maps with a spatial resolution limited to 100 km. For over sea areas, resolutions of 16 km can be reached by using radar satellite altimetry[1]. Higher spatial resolutions can only be obtained with airborne or ship-borne measurements. Until now, these surveys were carried out with relative sensors which only measure the variation of gravity and which suffer from drift. For a gravity survey, one needs thus to go regularly to a reference point where the gravity is known or where there is a static absolute gravimeter. Therefore, the use of a relative gravimeter has important operational constraints which increase the time and the cost of gravimetry survey and has measurement errors due to calibration and drift estimation. The use of an absolute gravimeter would thus be of great interest but until now these instruments can only work in static conditions. Only one feasibility study done with a modified FGL gravimeter on an aircraft[12] can be found in the literature.

Here, we present an absolute gravimeter able to measure gravity acceleration from a ship. This instrument is based on the acceleration measurement of a free falling gas of ultracold atoms with atom interferometry[13]. This technology has been developed since three decades and has allowed testing fundamental physics[14,15], measuring fundamental physics constants[16,17], and measuring with high precision gravity[18–21], gravity gradient[22,23], and rotation[24–26]. Most of these works consist in laboratory experiments but more and more atom interferometer are performed outside laboratory environment such as in a truck[27], an elevator[28], a zero-G airplane[29,30], a dropped tower[31], or a sounding rocket[32]. Concerning gravimeter, matter-wave sensors have now better or equal performances than classical absolute gravimeters[18–21] and have started to be commercialized. Some attempts have been made to use these technologies in a moving platform but, until now, only limited demonstrations could be found in the literature. In ref. [27], mobile gravity gradient measurements are reported in a truck moving at only 1 cm/s. In refs. [29,]30, acceleration measurements in a zero-G plane are reported but no gravity measurements were performed. In ref. [28], low-precision gravity measurements are reported in a moving elevator. In this work, we report a cold atom sensor performing absolute gravity measurements on a ship with a precision better than a usual calibrated spring gravimeter. This has been possible, thanks to several innovations such as the integration of a miniature atom sensor to a gyro-stabilized platform and the extension of the measurement range of the atom accelerometer by three orders of magnitude by combining it with a forced balanced accelerometer.

The gravimeter (see Fig. 1) is composed of an atom sensor which provides an absolute measurement of the acceleration, a gyro-stabilized platform which maintains the accelerometer aligned with the gravity acceleration despite angular movements of the ship, and systems which provide the lasers and microwaves needed for the atom sensor and perform data acquisition and processing.

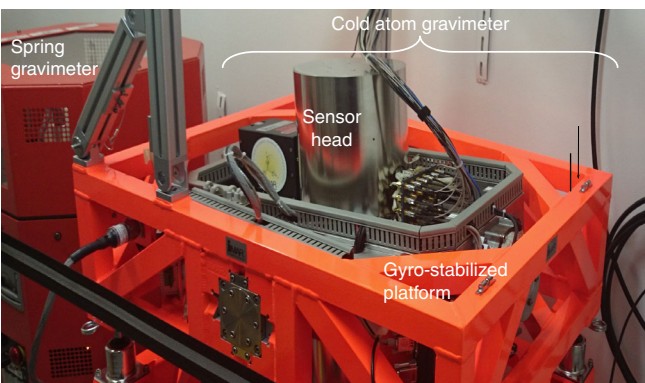

**Fig. 1** Cold atom gravimeter. Picture of the cold atom gravimeter installed in the Beautemps–Beaupré ship next to the spring gravimeter (KSS32M from Bodenseewerk)

## Results

**Description of the apparatus.** The atom accelerometer is similar to the one described in ref. [28]. The test mass is an ultracold gas of rubidium 87 atoms. It is produced from a magneto-optical trap loaded from a background vapor. After 20 ms of trap loading, a stage of optical molasses and a microwave selection, we obtain a cloud of one millions atoms in the magnetic field insensitive ground state $F = 1$, $m_F = 0$ and at a temperature of 1.9 μK. The acceleration of the free falling cloud of ultracold atoms is then measured by light pulse atom interferometry. For that, we use a Mach–Zehnder type atom interferometer consisting of a sequence of three equally spaced Raman laser pulses of duration 10, 20, and 10 μs which couple the two hyperfine ground states $F = 1$, $m_F = 0$ and $F = 2$, $m_F = 0$. The two counter-propagating laser beams addressing the Raman transition are obtained with a phase modulated laser at 6.8 GHz retro-reflected on a mirror[33]. In this interferometry sequence, the first pulse acts as a matter-wave beam splitter, the second one acts as a mirror, and the last one recombines the matter waves (see Fig. 2). The phase shift at the output of the interferometer is equal to $\phi = (k_{eff} \cdot a - \alpha)T^2$, where $k_{eff} \simeq 4\pi/\lambda$, $\lambda$ the laser wavelength, $T$ the time delay between Raman laser pulses, $a$ the acceleration of the atoms along the direction of the laser beam, and $\alpha$ the rate of the radiofrequency chirp applied to the 6.8 GHz frequency in order to compensate for the linearly increasing Doppler shift induced by the acceleration of the atoms. The interference signal is then obtained by measuring the population of atoms in the two hyperfine states corresponding to the two output ports of the interferometer. This measurement is obtained by a fluorescence method (see Methods section). The proportion $P$ of atoms in the state $F = 2$ after the atom interferometer sequence can be written as $P = P_m - C/2 \cos(\phi)$, where $P_m$ is the offset of the fringe and $C$ is the contrast which is typically equal to 0.3 for our sensor. Two different falling distances of 14 mm and 42 mm are possible in our sensor, leading to a maximum half interrogation time of, respectively, $T = 20$ ms and $T = 39$ ms and to a repetition rate of 10 Hz and 7 Hz. The long falling distance is used for static measurements and the short falling distance is more adapted to measurements in a moving vehicle.

The output $P$ of the atom sensor is proportional to the cosine of the acceleration with a period equal to $\lambda/2\ T^2$. For $T = 20$ ms, this period is equal to $10^{-3}$ m s$^{-2}$ and is small compared to the typical variations of acceleration in a moving vehicle. There is, therefore, an ambiguity to determine the acceleration from the measurement of the atom sensor. Many values of acceleration are possible for a given value of the output of the atom sensor. To overcome this limitation, we combine the atom sensor with a

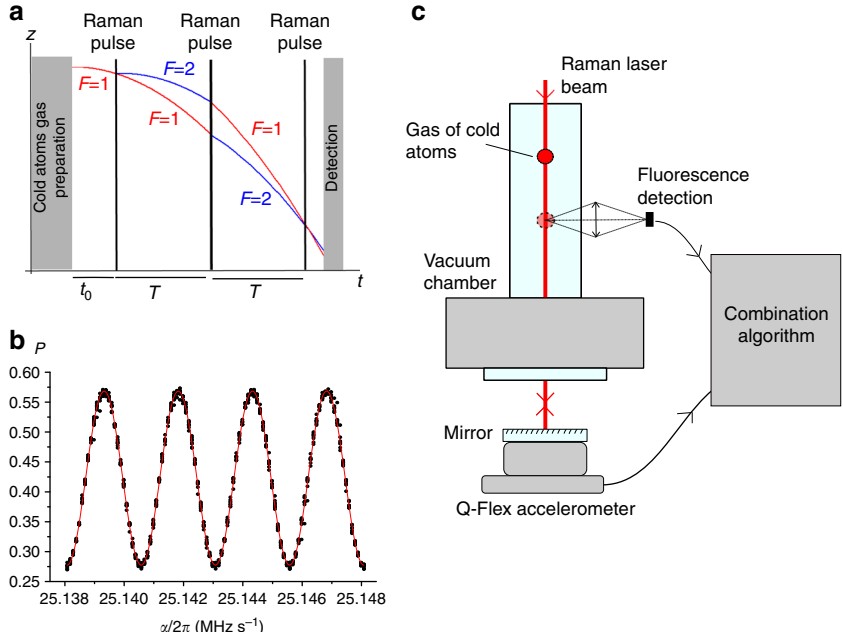

**Fig. 2** Principle of the atom accelerometer. **a** Temporal sequence. **b** Typical interference fringes acquired in static condition for $T = 20$ ms. **c** Setup of the cold atom accelerometer

classical accelerometer. This kind of method has already been successfully implemented in laboratory environment to measure the gravity acceleration[34,35] and in a free falling (zero-G) plane for acceleration measurements[29]. Here, we implement a robust combination scheme of the atom accelerometer with a force balanced accelerometer (Q- Flex from Honeywell). The classical accelerometer is used to give a first rough estimation of the acceleration in order to determine which value of acceleration corresponds to the signal of the atom sensor. The classical accelerometer is also used to measure the acceleration during the measurement dead times of the atom sensor which occur during the cold atoms preparation and during the detection. The filling of the measurement dead times is very important because vibrations at the repetition rate of the atoms sensors and its multiple can cause an important degradation of sensitivity due to aliasing effect. The complete description of the combination protocol is given in the Methods section.

This atom accelerometer has been implemented in a compact housing consisting of a cylinder of 22 cm diameter and 52 cm height. It is composed of a vacuum chamber made of glass in which the atoms are produced and interrogated, magnetic coils, optics for shaping all the laser beams and collecting the fluorescence of the atoms, two layers of mu-metal for shielding the external magnetic field, and classical accelerometers. This sensor is integrated in a two-axes stabilized gimbaled platform made by IMAR. The platform is stabilized using an integrated inertial measurement system and maintains the sensor head aligned with the gravity acceleration with a precision of 0.1 mrad. The lasers needed for the sensor are obtained using a compact frequency doubled telecom fiber bench[36] which is compatible with onboard environment.

**Evaluation of the gravimeter sensitivity and accuracy in static.** The gravimeter has been first characterized in static conditions. The measurement uncertainty has been estimated to 0.06 mGal (1 mGal = $10^{-5}$ m s$^{-2}$) with $T = 39$ ms using the 42 mm falling distance and to 0.17 mGal with $T = 20$ ms using the 14 mm falling distance. These uncertainties have been evaluated by analyzing the systematic effects affecting the sensor (see Methods section).

The estimation of the uncertainties has been confirmed by the comparison with an absolute A10 gravimeter. The difference between the two gravimeters has been measured equal to $0.01 \pm 0.06$ mGal for $T = 39$ ms and $0.09 \pm 0.17$ mGal for $T = 20$ ms. The measurement sensitivity of our gravimeter in static is equal to 0.8 mGal Hz$^{-1/2}$ limited by the sensitivity of the force balanced accelerometer.

**Marine gravity campaign.** The atom gravimeter has been then tested on a 85-m-long ship, BHO Beautemps-Beaupré, on which Shom (French hydrographic and oceanographic office) uses to realize gravity surveys for marine needs with a spring gravimeter KSS32M. The surveys with the atom gravimeter were done in North Atlantic Ocean to the west of French Brittany (see Fig. 3) in October 2015 and January 2016. During the surveys, we encountered particularly bad sea conditions with sea states from 4 to 6 corresponding to significant wave heights from 2 to 5 m. The gravimeter was subjected to vertical accelerations of frequency around 0.15 Hz and of amplitude ranging from 1 to 5 m s$^{-2}$ peak–peak. Only the 14 mm falling distance was used during the marine measurements and the half interrogation time chosen by the gravimeter algorithm was $T = 10$ ms or $T = 20$ ms. The data processing which allows to calculate the gravity anomaly from the acceleration measurement is described in Methods section. The results of the surveys are summarized in Table 1.

First, we measured the gravity along a profile crossing the continental slope where there is an important gravity signal. This is the calibration profile where all spring gravimeters of Shom are tested and we have, therefore, a very good knowledge of the gravity anomaly along the profile. The round trip measurements on this profile are shown in Fig. 4. A very good reproducibility is obtained between forward and backward measurements with a standard deviation on the difference equal to 0.5 mGal and a mean difference equal to 0.4 mGal. The measurements are also in good agreement with the reference data of Shom. The differences between the forward and backward measurements and the reference data have a standard deviation of 0.5 mGal and 0.3 mGal, respectively, and a mean value of −0.2 mGal and −0.6 mGal, respectively.

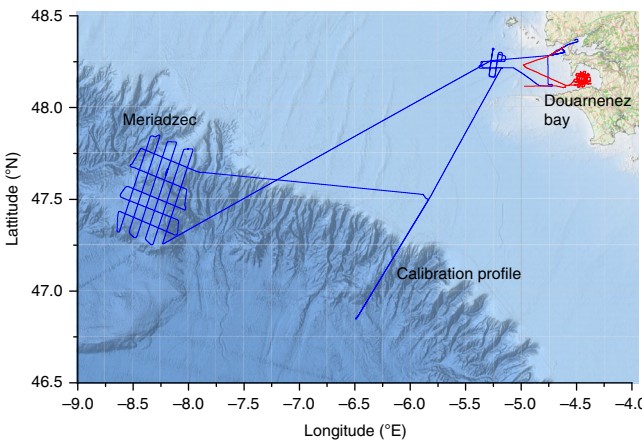

**Fig. 3** Map of the location of the gravity survey. Blue line: course of the ship during October 2015 survey. Red line: course of the ship during January 2016 survey. The background map is from EMODnet Bathymetry Consortium (2016): EMODnet Digital Bathymetry (DTM). https://doi.org/10.12770/c7b53704-999d-4721-b1a3-04ec60c87238

**Table 1 Gravimeters results and comparison**

| | | | Atom gravimeter | Spring gravimeter |
|---|---|---|---|---|
| Calibration profile (9 kn) | Forward–Backward | Mean | 0.4 | 1.8 |
| | | Std. | 0.5 | 0.9 |
| | Forward–Reference | Mean | −0.2 | 1.2 |
| | | Std. | 0.5 | 1.1 |
| | Backward–Reference | Mean | −0.6 | −0.5 |
| | | Std. | 0.3 | 0.6 |
| Meriadzec (9 kn) | Crossing points difference | Error | 0.9 | 1.0 |
| Douarnenez straight profiles (9 kn) | Forward–Backward | Mean | 0.1 | 0.1 |
| | | Std. | 0.2 | 0.8 |
| | Crossing points difference | Error | 0.4 | 1.0 |
| Douarnenez circular profiles (8 kn) | Crossing point difference with regular profile | Mean | −0.2 | 1.0 |
| | | Std. | 0.5 | 1.0 |
| Douarnenez circular profiles (11 kn) | Crossing point difference with regular profile | Mean | 0.3 | 2.8 |
| | | Std. | 0.6 | 2.9 |

All the values are in mGal. The gravity measurements were filtered with a spatial resolution of 0.8 km for the calibration profile and Douarnenez and 3 km for Meriadzec. The sea state was 6 for the calibration profile and Meriadzec and 4 for Douarnenez

Then, we measured the gravity along a grid on the Meriadzec terrace at the edge of the continental margin. The measurement error of this survey has been estimated to 0.9 mGal from the differences at the crossing points (the error is calculated by taking the standard deviation of the differences at the crossing points divided by $\sqrt{2}$[37]). A gravity model of the area has been established using the GMT software[38]. The extrapolation of the data was achieved using adjustable tension continuous curvature splines[39]. This model is compared to the one obtained by satellite altimetry[1] (see Fig. 5). The two models are similar but with a higher spatial resolution for the ship-borne model. The difference between the ship-borne measurements and the satellite model has a mean of 1.4 mGal and a standard deviation of 2.4 mGal in agreement with the estimated error of ship-borne measurements (0.9 mGal) and the estimated error of the satellite-borne model (2 mGal).

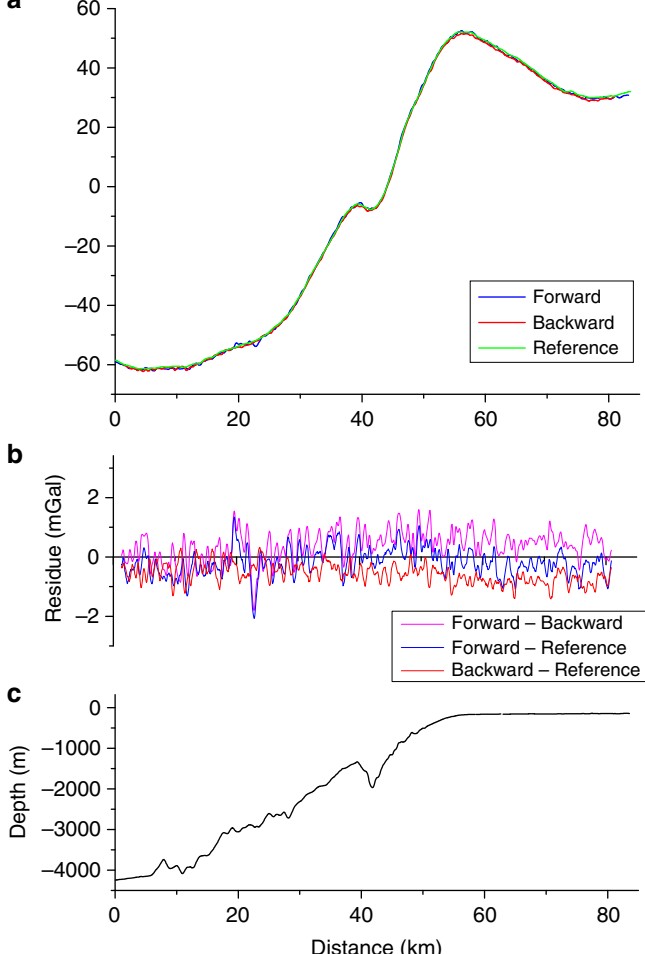

**Fig. 4** Absolute gravity measurements along the calibration profile. **a** Free air gravity anomaly measured by the atom gravimeter (forward and backward) and free gravity anomaly from Shom reference data. **b** Difference between the forward and the backward measurements and the reference data of Shom. **c** Depth along the calibration profile

Finally in the Douarnenez bay, we measured gravity twice along 8 straight lines (round trip) with a velocity of 9 kn and along two circular lines of radius 1.9 km and 3.3 km with two different velocities of 8 kn and 11 kn for each circle. The measurements along the circular profile were done in order to investigate the precision of gravity measurements during the turning of the ship. For straight profiles, the difference between forward and backward has a mean value of 0.1 mGal and a standard deviation of 0.2 mGal. From the differences at the crossing points, we estimate an error of 0.4 mGal. The precision in circular profile has been estimated by using the difference at the crossing points with the linear profiles. For the circles at 8 kn, the differences have a mean value of −0.2 mGal and a standard deviation of 0.5 mGal. For the circles at 11 kn, the differences have a mean value of 0.3 mGal and a standard deviation of 0.6 mGal. With the gravity measurements along straight and circular lines, we established a gravity model of the area (see Fig. 6) by using the same method as Meriadzec terrace. This model is compared to the model obtained by satellite altimetry[1]. We see clearly here that the satellite model does not reproduce the gravity signal deduced from the ship-borne measurement. The satellite model has an offset of 7.2 mGal and a very low spatial resolution. This highlights the fact that satellite altimetry gravity model is not precise in coastal areas and that ship-borne or airborne

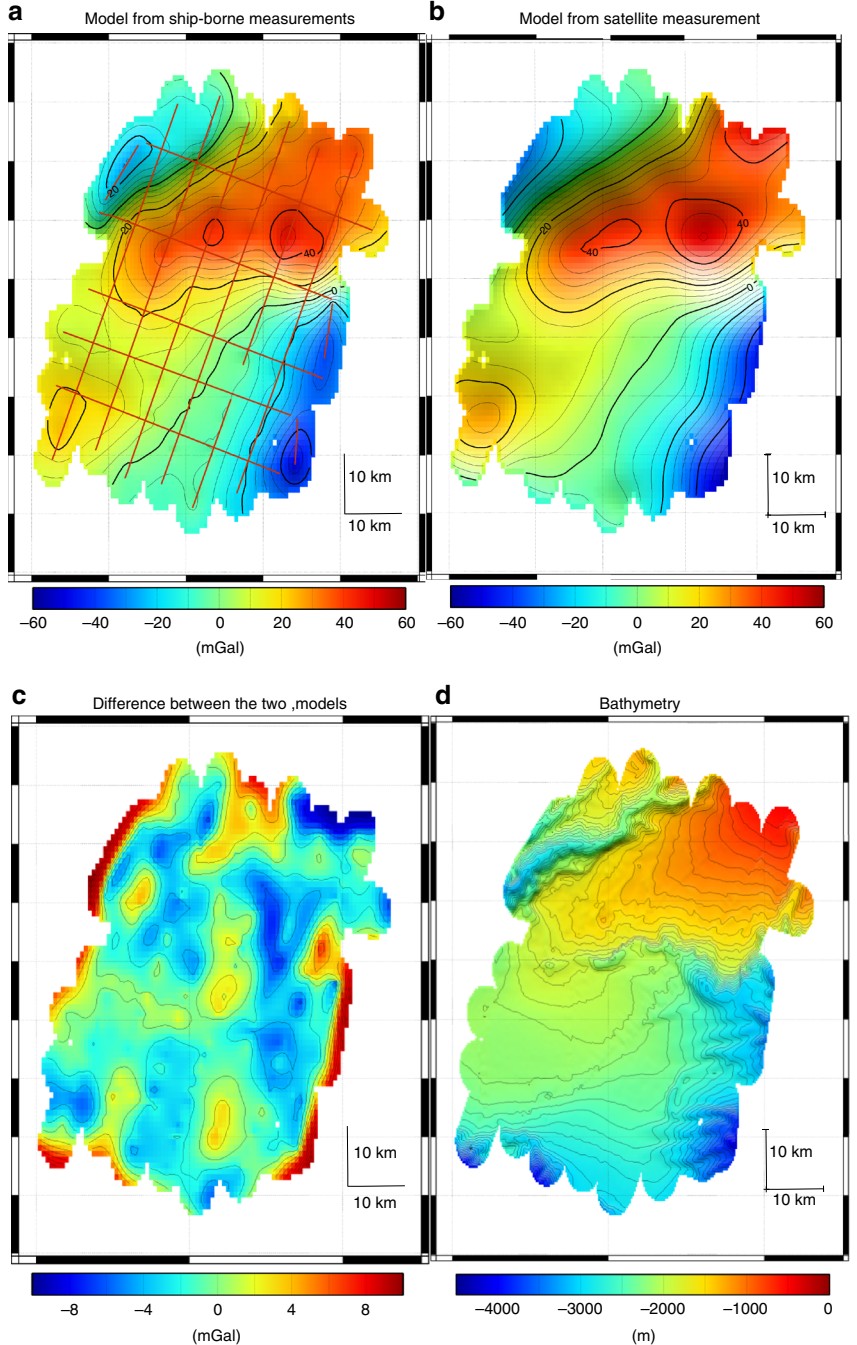

**Fig. 5** Gravity anomaly model of Meriadzec terrace. **a** Model obtained from ship-borne atom gravimeter measurements. The red lines are the profiles on which the gravity was measured. **b** Model obtained from satellite measurements[1] (Sandwell v24). **c** Difference between the ship-borne model and the satellite model. The important differences on the left and right border are due to the extrapolation procedure of the ship-borne gravity measurements. **d** Bathymetry for comparison with gravity anomaly

measurements are essential for gravity measurements in these areas.

The precision of the atom gravimeter has been compared with a relative spring gravimeter KSS32M (see Table 1). The two gravimeters were placed next to each other in the ship (see Fig. 1) and the data processing was the same, except for the calibration process for the spring gravimeter. Systematically better precision is obtained for the atom gravimeter with an improvement factor up to 5. This improved precision can be attributed to the removal of calibration error for the absolute atom gravimeter, to the intrinsic better precision of the cold atom sensor, and to the good quality of the gyro-stabilized platform. The last point is clearly

visible with the measurements in circular profile in which the platform of the relative gravimeter is responsible of performance degradations.

## Discussion

In conclusion, we demonstrated sub-mGal ship-borne gravity measurements with a matter-wave sensor. This technology has allowed us to obtain absolute gravity measurements from a ship and to improve the precision compared to a conventional spring gravimeter. The atom gravimeter could address other carrier like aircraft or underwater vehicle and thus offers a development in

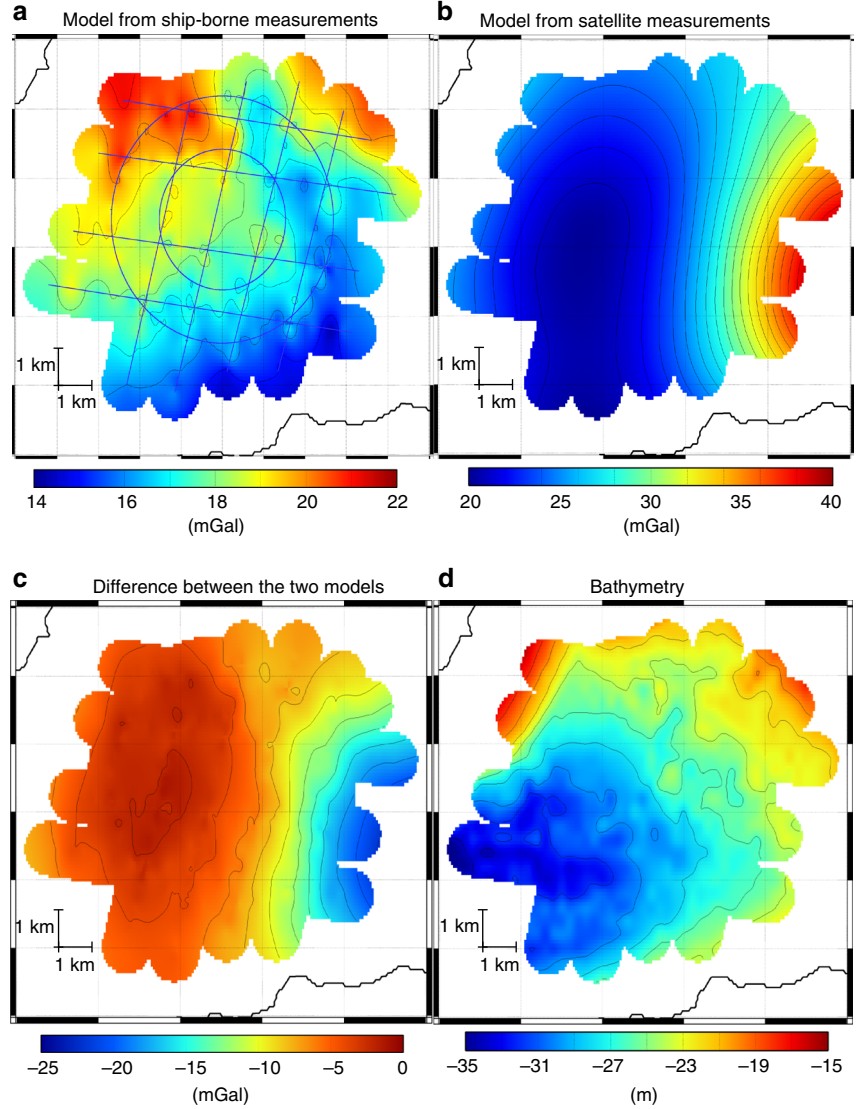

**Fig. 6** Gravity anomaly model of Douarnenez bay. **a** Model obtained from ship-borne atom gravimeter measurements. The blue lines are the profiles on which the gravity was measured. **b** Model obtained from satellite measurements[1] (Sandwell v24). **c** Difference between the ship-borne model and the satellite model. **d** Bathymetry for comparison with gravity anomaly

onboard gravimetry. Our results support also the development of matter-wave sensor for measuring the Earth gravity field from space[40]. Finally, the demonstration of absolute acceleration measurements in dynamic opens the way to the next generation of inertial sensors (accelerometer and gyroscope) able to make absolute measurement in a dynamic environment. We can therefore imagine in the future absolute Inertial Measurement Units which do not drift and do not need to be calibrated.

## Methods

**Detection method**. The proportion of atoms in the state $F = 2$ and $F = 1$ is measured by collecting the fluorescence of the atoms illuminated with three pulses of the vertical retro-reflected laser beam of duration 1.5, 0.5, and 1.5 ms. The first pulse and the last pulse are resonant with the $F = 2 \rightarrow F' = 3$ transition and give a fluorescence signal proportional to the number of atoms in the state $F = 2$. During these laser pulses, the atoms in the state $F = 2$ are pushed away from the detection zone by the radiation pressure force of the detection beam. The middle pulse is resonant with the $F = 1 \rightarrow F' = 2$ transition and transfers the atoms from the state $F = 1$ to the state $F = 2$. The fluorescence signal during the first pulse is thus proportional to the number of atoms initially in the state $F = 2$ and the fluorescence signal during the third pulse is proportional to the number of atoms initially in the state $F = 1$. An additional laser pulse of 1.5 ms duration is applied in order to acquire the background.

**Protocol of atom and force balanced accelerometers combination**. The acceleration is deduced from the atom sensor signal $P$ by using the following equation:

$$a_{at} = \frac{s \times a\cos\left(2\frac{P_m - P}{C}\right) + 2\pi \times n}{k_{eff} T^2} + \frac{\alpha}{k_{eff}},$$

where $s = \pm 1$ and $n$ is an integer. The sign $s$ and the integer n are determined to obtain the closest acceleration from the estimated acceleration $a_{at/FB}$ given by the forced balanced accelerometer. $a_{at/FB}$ is obtained by convoluting the measurement of the forced balanced accelerometer $a_{FB}$ by the response function of the atom accelerometer $h_{at}$:

$$a_{at/FB} = \int_{-T}^{T} a_{FB}(t_\pi + t) \times h_{at}(t) dt,$$

where $t_\pi$ is the instant of the middle laser pulse of the atom interferometer sequence and $h_{at}$ is the atom response function given by the triangle-like function:

$$h_{at}(t) = \quad \frac{T-t}{T^2} \text{ if } t \in [0, T]$$

$$= \frac{T+t}{T^2} \text{ if } t \in [-T, 0]$$

The difference between $a_{at}$ and $a_{at/FB}$ is used to estimate continuously the bias of the force balanced accelerometer. The value of $P_m$ and $C$ are estimated

**Table 2 Bias and uncertainties of the main systematic effects affecting the cold atom gravimeter for two different interrogation times**

| | T = 20 ms | | T = 39 ms | |
|---|---|---|---|---|
| | Bias | Uncert. | Bias | Uncert. |
| Light shift second order | 0.428 | 0.027 | 0.076 | 0.024 |
| Additional laser lines | 0.261 | 0.160 | 0.638 | 0.038 |
| Coriolis effect | 0 | 0.030 | 0 | 0.030 |
| Wavefront curvature | 0 | 0.026 | 0 | 0.026 |
| Variation of Raman laser frequency | 0.110 | 0.030 | 0.029 | 0.008 |
| Total | 0.80 | 0.17 | 0.74 | 0.06 |

The last line (total) is the sum of all the systematic effects and represents the bias and the uncertainty of the gravity measurements

continuously by assuming that the phase of the interferometer is randomly distributed and by measuring the mean value and the standard deviation of the atom sensor signal.

The radiofrequency chirp $\alpha$ that we apply at each measurement cycle is equal to:

$$\alpha = \pm 1 \left( k_{eff} \times a_{prev} + \frac{rnd(2\pi)}{T^2} \right),$$

where $rnd(2\pi)$ is a random number with a uniform distribution between 0 and $2\pi$, $a_{prev}$ is the acceleration measured at the previous measurement cycle. This choice of the radiofrequency chirp $\alpha$ ensures to keep the Raman laser in resonance with the atoms during their fall and to have a phase of the interferometer randomly distributed between 0 and $2\pi$ regardless of variations of acceleration. The sign of the chirp $\alpha$ is also changed in every measurement cycle to cancel some systematic effects.

The continuous acceleration measurement is obtained by filling the dead time of the atom accelerometer with the measurement of the force balanced accelerometer:

$$a_{cont} = a_{at} + a_{cont/FB} - a_{at/FB},$$

where $a_{cont/FB}$ is the mean acceleration measured by the force balanced accelerometer over one complete measurement cycle.

This protocol allows having a continuous and absolute measurement of the acceleration with a dynamical range compatible with onboard applications. This method is working well if the error of the estimated acceleration by the classical accelerometer is much smaller than the period of the atom accelerometer signal $\lambda/2T^2$. The main sources of this error are the different localization of the measurement points, the misalignment between the accelerometers, the bias of the classical accelerometer and the uncertainty of the scale factor, and the transfer function of the classical accelerometer. These errors have been minimized in our instrument; however, in hard dynamical environments or when the bias of the classical accelerometer is not yet determined, the error is too large for the biggest interrogation time ($T = 20$ ms). To overcome this limitation, an automatic determination of the optimum interrogation time is implemented. The algorithm chooses between the following values of $T$: 2.5, 5, 10, or 20 ms depending on the measured difference between the atom accelerometer and the estimation of the classical accelerometer. Therefore, at the start of the instrument when the bias of the classical accelerometer is unknown, the algorithm starts with $T = 2.5$ ms compatible with the initial bias of the classical accelerometer. The algorithm increases then progressively $T$ to its maximum value of $T = 20$ ms as the bias of the classical accelerometer becomes estimated.

**Error budget**. The main systematic effects which limit the accuracy of the gravimeter are listed in Table 2. The biases caused by the non-homogeneity of the magnetic field and by the first-order light shift are not reported in this table because they are canceled with our protocol of changing the sign of $k_{eff}$ and thus the sign of $k_{eff}$ at each measurement cycle. The second-order light shift[41,42] has been calibrated by measuring gravity vs. the power of the Raman laser. The generation of the Raman laser by modulation produces additional laser lines which are responsible of a bias. This effect was calibrated by using the method described in ref.[33] The uncertainty given by the Coriolis effect with the Earth rotation has been estimated by taking an uncertainty on the transverse velocity of the atoms equal to 3 mm s$^{-1}$. The uncertainty caused by the wavefront curvature of the Raman laser beams[43,44] has been evaluated, thanks to the estimation of the wavefront deformation induced by our optics. Due to the limited bandwidth of the laser frequency lock, the Raman laser frequency is not perfectly the same for the three laser pulses

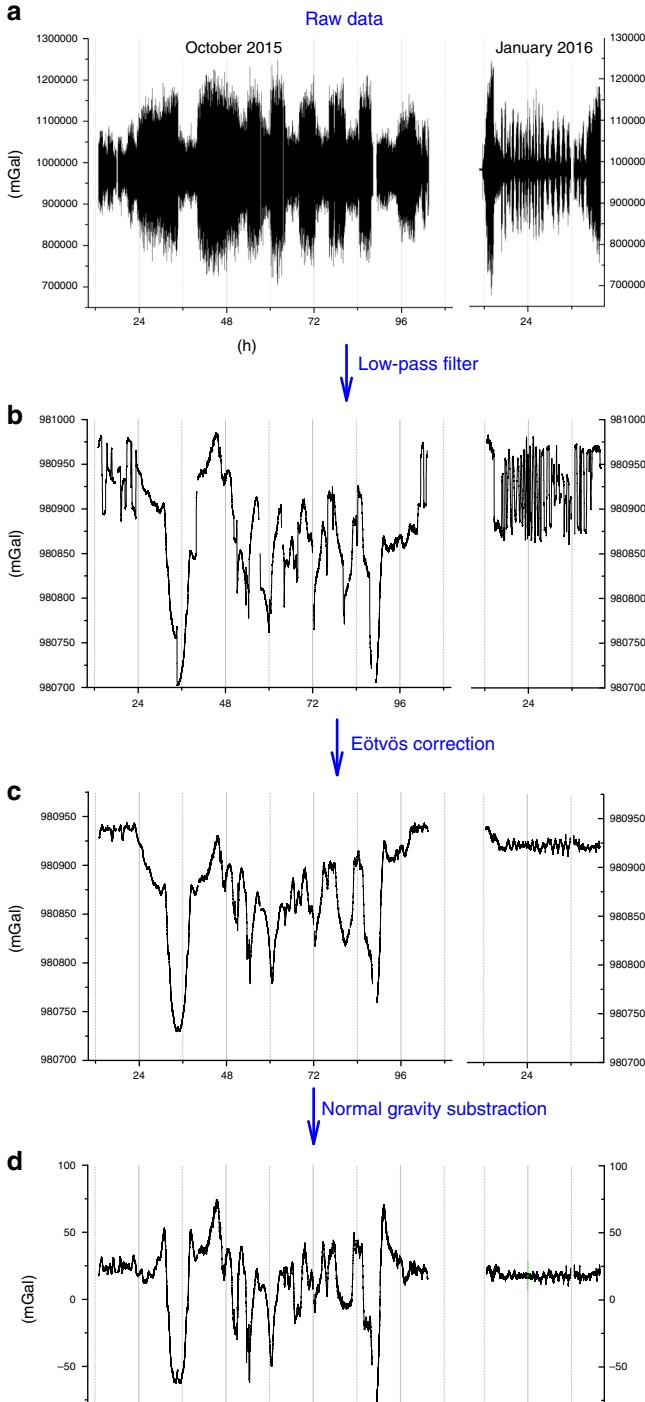

**Fig. 7** Illustration of data processing during the gravimetric survey of October 2015 and January 2016. **a** Raw data ($g + a_{ship} + a_{Eötvös}$). **b** Raw data low-pass filtered with a Bessel 4th order filter of time constant 130 s ($g + a_{Eötvös}$). **c** Gravity measurements ($g$). **d** Gravity anomaly measurements ($g - g_{norm}$)

and causes a bias equal to

$$\Delta a = \frac{(2\omega_{L2} - \omega_{L1} - \omega_{L3}) \times t_d}{|k_{eff}| \times T^2},$$

where $\omega_{Li}$ is the Raman frequency laser at the $i$th pulse and $t_d = 2 L/c$ with $L$ the distance between the atoms and the retro-reflection mirror. This bias has been estimated by measuring the laser frequency for each pulse.

**Data processing**. The gravimeter is measuring the acceleration along the gravity acceleration with the ship as a reference frame. Therefore, in addition to the gravity acceleration $g$, the sensor is measuring the vertical acceleration of the ship $a_{ship}$ and the Coriolis acceleration $a_{Eötvös}$ (Eötvös effect) due to the coupling of the Earth rotation and the ship velocity on Earth surface. To deduce the gravity anomaly from the measurements, we apply the following data processing (see Fig. 7). Since the ship remains on average at the same elevation, the vertical acceleration of the boat $a_{ship}$ can be filtered by applying a low-pass filter (Bessel 4th order). The choice of the filter time constant depends on the sea conditions and is a trade off between the spatial resolution and the filtering of vertical acceleration of the ship. The Eötvös acceleration is given by:

$$a_{Eötvös} = 2 \times \Omega_{Earth} \times v \times \sin(\phi_{head}) \times \cos(\phi_{lat}) + \frac{v^2}{R_{Earth}},$$

where $v$ is the ship velocity, $\phi_{head}$ is the ship heading, $\phi_{lat}$ is the latitude, $\Omega_{Earth}$ is the Earth rotation rate, and $R_{Earth}$ is the Earth radius. The data are corrected from this effect by using navigation data coming from the inertial navigation system of the ship. Finally, the gravity anomaly is obtained by subtracting to the data the normal gravity model[45] ($g_{norm}$).

**Data availability**. The data used in this manuscript are available from the corresponding author upon reasonable request.

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

## Acknowledgements

This work was funded by the French Defense Agency (DGA). We thank the crews of the Beautemps Beaupré ship for the marine test of the gravimeter. We thank G. Delachienne for the installation of the gravimeter in the ship. We thank S. Lucas for the gravity measurements in ONERA laboratory with an A10 gravimeter. We thank Y. Moysan for fruitful discussions on gyro-stabilized platforms. We thank the DGA team who had followed the project for fruitful discussions.

## Author contributions

Y.B. designed, built, and tested the gravimeter. N.Z., M.C., and A.Br. contributed to the design of the gravimeter. C.B. contributed to the electronic of the gravimeter. N.Z., A.Bo., A.Br., and Y.B. designed and built the laser system. N.Z. and A.Br. designed and built the microwave system. The automation and the programming were done by A.Br. Y.B., C.B., A.Br., D.R., and M.F.L. participated in the marine campaigns. The data processing and analysis of the gravity measurements were done by Y.B., A.Br., M.F.L., and D.R. M.F.L.

supervised the project in Shom. Y.B. wrote the manuscript with contributions from all authors.

## Additional information

**Competing interests:** The authors declare no competing financial interests.

