## [Peer Review File · Nature Communications]

Reviewers' comments:

Reviewer #1 (Remarks to the Author):

In this manuscript, the authors describe the use of an atom interferometer gravimeter to make gravity maps on a mobile platform - a ship in the at-times-heavy seas of the North Atlantic. This is a nice demonstration of the power and utility of these devices, and an important step towards commercialization, but I don't feel that there is enough novel contributions here to warrant publication in Nature Comm.

Atom interferometer gravimeters are over 20 years old, and the use on mobile platforms is about ten years old. The authors don't provide any references from the Kasevich group, but that group performed mobile platform tests of a compact and high-bandwidth gravity gradiometer about a decade ago. This work is described in the published thesis of Grant Biederman, Stanford University (2008), "Gravity Tests, Differential Accelerometry, ..." Certainly a ship-based platform is vastly more challenging than the truck-based tests performed by Kasevich, et al., but I see this as an incremental step. Furthermore, it is my understanding that the community has a preference for gradiometry, to circumvent the Equivalence Principle and distinguish platform accelerations from gravitational signatures.

Other than demonstration in a more challenging environment, there is not much new contribution here. The hybrid measurement scheme to avoid acceleration ambiguities has been discussed elsewhere, and the technical details of operation seem fairly standard. Thus, I recommend that a more specialized navigation/geodesy journal would be more appropriate for this manuscript.

Reviewer #2 (Remarks to the Author):

The authors describe the use of a Rb atom interferometer in the by now 'classical' Raman/Kasevich/Chu configuration ($\pi/2 - \pi - \pi/2$) for on-ship gravity measurements. It is interesting to see that g-measurements can be made on a boat with 10 knots speed. This is a technical feat.

On the side of scientific results it is not clear if this meets the requirements of Nature Communications. On the purely scientific result side, I would rather see the manuscript rather in a more technical journal.

It has the extra appeal of being interdisciplinary, combining atom optics with marine geodesy but without really making full use of this scientific potential yet.

- The atom interferometer concept as such as first been published in 1991 and the dozens of times in high level journals, also for gravity measurements. The authors refer to it as the first interferometer on a moving platform. But do they count an airplane in free fall as moving? This was published in Nature Communications a few years ago: Geiger, R. et al. Detecting inertial effects with airborne matter-wave interferometry. Nat Commun 2, 474, doi:10.1038/ncomms1479 (2011). What about recent sounding rocket tests? In particular since the authors conclude they would suggest their device for "the development of matter wave sensor for measuring the Earth gravity field from space". Has the state of the art not yet been pushed further than recognized here, by the MAIUS mission? I would also be surprised if there had not been any measurements on trucks? This would be the first and by far cheapest test of any such system, since very few groups own a large ship, a plane or a rocket.

- They reach a sensitivity and accuracy of 10^{-5} m/s². Transportable (not moving) commercial standards perform 3 orders of magnitude better (also in absolute terms). Static machines even seven orders of magnitude or more. Which problem exactly can they address on that level of accuracy? What are the demands of prospection for natural resources and how would they meet them?

- They compare their instrument to spring gravimeters, but for a general audience I would see the need to compare also to a larger class of gravimeters, such as falling corner cubes and squids etc.

Some general comments

- The overall structure of the manuscript is clear and focused.
- Figure 1 looks nice and orange but information is sparse. What I learn from it: they have built the device, it is bulky but OK sized and orange. I would look for more scientific info rather than a photo. The most crucial part seems to be the isolation platform, which is nowhere explained in detail
- Figure 3 looks nicely blue, but again the information is limited. Unless the location is important – and local g-measurements are correlated with concrete scientific insight – the figure only tells the reader that the boat has indeed cruised quite a bit and covered rifts where g must vary. But to assess that, it would be valuable to know the depth profile. This would be the only interesting information and this is missing. The reader may find this themselves in Google Earth.
- Table 1 is interesting, but not used to extract any scientific claim. The discrepancies between atom and spring gravimeter are not further explained. It is not only about the absolute values but even the signs differ in some locations. Would it make sense to correlate the table with depth profiles in Figure 3 to provide both with some geographical and physical meaning? In many cases the Standard Deviation is as high as the mean value. How much and how reliable data can you then extract from it?
- Figure 5 and 6 compare the atom interferometry data with satellite measurements. The resolution in the atom interferometry data seems to be somewhat better and the profiles seem to disagree sometimes somewhat. But how to judge which system is more correct? What do we learn?
- The data seem to be more accurate for the atom interferometer compared to the spring gravimeter. As one of the reasons the authors mention the removal of the calibration error in the atom system – which is an important and very positive point in this design - but also the additional gyro-platform. What if this platform were combined with the spring sensor? Would the classical and the quantum system then compete again?
- To my taste the word “hybridized” is used excessively without really adding information. It just means they combine a classical accelerometer with the atom interferometer. This must have been done in any atom interferometer in the world by now (and properly referred to by the authors).
- I miss details of the gyro-stabilized platform. This is important as the measurements would not work without it.
- The authors refer to two different falling times as “possible” (20 and 39 ms). Why should there be no continuous transition between them?
- Is the length of the ship (85 m) relevant to the measurement? I would not see how? Like some of the pictures, this info seems to be more relevant in a talk to the general public. But of course it does not harm.
- As to the sentence: “on which Shom uses to realize gravity surveys for marine needs with a spring gravimeter KSS32M.” Outside of France “Shom” is probably less known and not everyone reads footnotes in the first run. Please explain all acronyms and names in the main text.
- The authors refer to “sea states from 4 to 6”: Not being a mariner, Google told me the definition of a sea state, but why puzzle the layman and general audience with sailor’s yarn instead of going straight for the qualitative and quantitative statements for scientists: “rough to high waves with wind speeds between 16-30 knots and wave heights between 5 and 20 feet.”
- Equation (1) is not explicitly referred to, does not need to be a displayed equation, anyhow, and

could be integrated in the section below.

- On page one they refer to the radiofrequency chirp $a/2n$ without first explaining what the concept of the measurement is. I suppose, in the atom interferometry community everybody will know it. Then you need not even define it. For all others, shifting it a few lines down, would be helpful. The units ($2n$) are irrelevant in the text and sufficient in the graph.
- The reader is given the millimeter extensions of the housing of the classical gravimeter (215 x 517 mm): I was surprised to see this, because it appears rather vain information, or do I overlook anything important here? Of course, the overall size and weight of the entire "hybridized system" could be of interest (but even then decimeter accuracy suffices) to see how transportable it is.
- The acceleration "*a*" should be written in italics, not to mix it with the indefinite article
- Spelling and grammar still require strong revision in many places:
 - Articles or plural endings are often missing - too often to list them all here. That may be quickly corrected.
 - attempts are usually "made" not "done"
 - in two instances the authors use the phrase we report ... to refer to work by other groups. This may be misleading, even though they properly cite the others. At least I would expect own work to follow such a phrase.
 - the continuous form "is corresponding" is not appropriate for something that is generally true or repeated etc. A native English reader will sort this out in a few minutes.
 - the citations are dominantly taken from the wide geodesy community. Since the paper does not report any new geodetic information but rather a technological advance in making an atom interferometer robust and isolated for a rough sea, one should probably add citations from the atom interferometry community, too. I was surprised too see almost only french groups cited, while US groups, German, Australian, British and more recently also Chinese groups have been pushing the field enormously. This should be taken into account.

Point-by-point response to the referees' comments

Reviewer #1 (Remarks to the Author):

In this manuscript, the authors describe the use of an atom interferometer gravimeter to make gravity maps on a mobile platform - a ship in the at-times-heavy seas of the North Atlantic. This is a nice demonstration of the power and utility of these devices, and an important step towards commercialization, but I don't feel that there is enough novel contributions here to warrant publication in Nature Comm.

Atom interferometer gravimeters are over 20 years old, and the use on mobile platforms is about ten years old. The authors don't provide any references from the Kasevich group, but that group performed mobile platform tests of a compact and high-bandwidth gravity gradiometer about a decade ago. This work is described in the published thesis of Grant Biederman, Stanford University (2008), "Gravity Tests, Differential Accelerometry, ..." Certainly a ship-based platform is vastly more challenging than the truck-based tests performed by Kasevich, et al., but I see this as an incremental step. Furthermore, it is my understanding that the community has a preference for gradiometry, to circumvent the Equivalence Principle and distinguish platform accelerations from gravitational signatures.

Response :

The reference (thesis of Grant Biederman, Stanford University (2008), "Gravity Tests, Differential Accelerometry, ...") cited by one reviewer is concerning a gravity gradiometer and not a gravimeter that is why we did not cite it in our paper focused on gravimetry . Moreover, the dynamical measurements were done with a truck moving at only 1 cm/s which is far from a real dynamical environment. To our point of view, it is not an incremental step to have a gravimeter working in ship with wave height from 2 m to 5 m compared to a gradiometer in a truck moving at only 1 cm/s. Moreover, concerning vibration and platform acceleration, a gravimeter is more challenging than a gradiometer which it is not sensitive to acceleration and vibration. For the community, gradiometer and gravimeter are complementary instruments; gravimeter is more precise for long wavelength gravity anomaly and gradiometer for short wavelength gravity anomaly.

Other than demonstration in a more challenging environment, there is not much new contribution here. The hybrid measurement scheme to avoid acceleration ambiguities has been discussed elsewhere, and the technical details of operation seem fairly standard. Thus, I recommend that a more specialized navigation/geodesy journal would be more appropriate for this manuscript.

Response :

Our work reports for the first time mobile gravity measurement with a matter wave sensor in a real operating environment. Moreover, compared to a state of the art mobile gravimeter, improved precision have been obtained. To our point of view, it is an important contribution here.

It is true that the principle of the hybrid measurement is known. But we implement a new algorithm that allows obtaining precise gravity measurements despite high level of acceleration variation. The details of this algorithm are described in the method section.

Reviewer #2 (Remarks to the Author):

The authors describe the use of a Rb atom interferometer in the by now ‘classical’ Raman/Kasevich/Chu configuration ($\pi/2 - \pi - \pi/2$) for on-ship gravity measurements. It is interesting to see that g-measurements can be made on a boat with 10 knots speed. This is a technical feat.

On the side of scientific results it is not clear if this meets the requirements of Nature Communications. On the purely scientific result side, I would rather see the manuscript rather in a more technical journal.

It has the extra appeal of being interdisciplinary, combining atom optics with marine geodesy but without really making full use of this scientific potential yet.

- The atom interferometer concept as such as first been published in 1991 and the dozens of times in high level journals, also for gravity measurements. The authors refer to it as the first interferometer on a moving platform. But do they count an airplane in free fall as moving? This was published in Nature Communications a few years ago: Geiger, R. et al. Detecting inertial effects with airborne matter-wave interferometry. Nat Commun 2, 474, doi:10.1038/ncomms1479 (2011). What about recent sounding rocket tests? In particular since the authors conclude they would suggest their device for “the development of matter wave sensor for measuring the Earth gravity field from space”. Has the state of the art not yet been pushed further than recognized here, by the MAIUS mission? I would also be surprised if there had not been any measurements on trucks? This would be the first and by far cheapest test of any such system, since very few groups own a large ship, a plane or a rocket.

Response :

We refer our work as the first gravity measurement with a matter wave sensor in a mobile platform. In the reference: "Geiger, R. et al. Detecting inertial effects with airborne matter-wave interferometry. Nat Commun 2, 474, doi:10.1038/ncomms1479 (2011)", only acceleration measurements in the zero g plane are reported, no gravity measurements are reported. Concerning the sounding rocket with the Maius mission, the work has not been published yet but it seems from the web page (<https://www.research-in-germany.org/en/research-landscape/news/2017/01/2017-01-23-maius-1---first-bose-einstein-condensate-generated-in-space.html>) that there is only the demonstration of matter wave interference in space with a BEC and no gravity measurements were achieved. To our knowledge, no cold atom gravimeter measurement in a truck is reported in the literature.

- They reach a sensitivity and accuracy of 10^{-5} m/s^2 . Transportable (not moving) commercial standards perform 3 orders of magnitude better (also in absolute terms). Static machines even seven orders of magnitude or more. Which problem exactly can they address on that level of accuracy? What are the demands of prospection for natural resources and how would they meet them?

Response :

Existing mobile gravimeter have all accuracy at the level of 10^{-5} m/s^2 (1 mGal). That level of accuracy allows addressing many problems in geophysics (ice melting issue, volcano, ground water variation ...) and oil, gas and mineral exploration.

- They compare their instrument to spring gravimeters, but for a general audience I would see the need to compare also to a larger class of gravimeters, such as falling corner cubes and squids etc.

Response :

Spring gravimeter is the reference gravimeter for mobile gravimetry. That is why the cold atom gravimeter is compared to spring gravimeter in our manuscript focused on mobile gravimetry. Falling corner cubes and superconducting gravimeter are used for static applications.

Some general comments

- The overall structure of the manuscript is clear and focused.
- Figure 1 looks nice and orange but information is sparse. What I learn from it: they have built the device, it is bulky but OK sized and orange. I would look for more scientific info rather than a photo. The most crucial part seems to be the isolation platform, which is nowhere explained in detail

Response :

The picture gives an overview of the cold atom gravimeter and also allows comparing its size with the spring gravimeter.

- Figure 3 looks nicely blue, but again the information is limited. Unless the location is important – and local g-measurements are correlated with concrete scientific insight – the figure only tells the reader that the boat has indeed cruised quite a bit and covered rifts where g must vary. But to assess that, it would be valuable to know the depth profile. This would be the only interesting information and this is missing. The reader may find this themselves in Google Earth.

Response :

Figure 3 gives an overview of the locations of the gravimetry survey. On this map, one can see clearly the continental slope responsible of the gravity anomaly measured in the survey. The map allows also locating the survey compare to the coast. This is important for the comparison with the gravity model of the satellite.

- Table 1 is interesting, but not used to extract any scientific claim. The discrepancies between atom and spring gravimeter are not further explained. It is not only about the absolute values but even the signs differ in some locations. Would it make sense to correlate the table with depth profiles in Figure 3 to provide both with some geographical and physical meaning? In many cases the Standard Deviation is as high as the mean value. How much and how reliable data can you then extract from it?

Response :

Table 1 gives information about the precision of the cold atom gravimeter in different situation. It allows also comparing the performance of the cold atom gravimeter and the spring gravimeter. The gravity anomaly depends on the depth profile but also on the density of the underground. At our level of precision, the comparison with the depth profile does not give any supplementary information.

- Figure 5 and 6 compare the atom interferometry data with satellite measurements. The resolution in the atom interferometry data seems to be somewhat better and the profiles seem to disagree sometimes somewhat. But how to judge which system is more correct? What do we learn?

Response :

On Figure 5, this comparison validates our gravity measurements. The two measurements agree within their uncertainties. Figure 6 illustrates the fact that satellite altimetry gravity model is not precise in coastal area and that ship borne and airborne measurements are essential in these areas.

- The data seem to be more accurate for the atom interferometer compared to the spring gravimeter. As one of the reasons the authors mention the removal of the calibration error in the atom system – which is an important and very positive point in this design - but also the additional gyro-platform. What if this platform were combined with the spring sensor? Would the classical and the quantum system then compete again?

Response :

The main advantage of the quantum system is its propriety of being absolute i.e no calibration and no drift. This makes a gravimetry survey cheaper, faster and easier with no need to go regularly to a reference point to calibrate the sensor. Concerning the improved precision of the cold atom sensor, it is difficult now to estimate which part comes from the removal of calibration error, the precision of the sensor itself or the quality of the gyro-stabilized platform.

- To my taste the word “hybridized” is used excessively without really adding information. It just means they combine a classical accelerometer with the atom interferometer. This must have been done in any atom interferometer in the world by now (and properly referred to by the authors).

Response : "hybridized" is a term used in the community (see ref. 22.). In the literature, only few hybridized atom interferometer are reported.

- I miss details of the gyro-stabilized platform. This is important as the measurements would not work without it.

Response : The information of the gyro-stabilized platform are given on p. 2. We do not have more details because it is a commercial product.

- The authors refer to two different falling times as “possible” (20 and 39 ms). Why should there be no continuous transition between them?

Response : The two different falling times correspond to two different detection set-up that is why there is no continuous transition between them.

- Is the length of the ship (85 m) relevant to the measurement ? I would not see how? Like some of the pictures, this info seems to be more relevant in a talk to the general public. But of course it does not harm.

Response : The length of the ship is an important parameter because it is related to the movement of the boat for a given sea state. A small ship will move more than a big ship for a given sea state.

- As to the sentence: “on which Shom uses to realize gravity surveys for marine needs with a spring gravimeter KSS32M.” Outside of France “Shom” is probably less known and not everyone reads footnotes in the first run.

Please explain all acronyms and names in the main text.

Response : correction done in the revised version of the manuscript.

- The authors refer to “sea states from 4 to 6”: Not being a mariner, Google told me the definition of a sea state, but why puzzle the layman and general audience with sailor’s yarn

instead of going straight for the qualitative and quantitative statements for scientists: “rough to high waves with wind speeds between 16-30 knots and wave heights between 5 and 20 feet.”

Response : In the manuscript, the correspondence wave height and sea state is given.

- Equation (1) is not explicitly referred to, does not need to be a displayed equation, anyhow, and could be integrated in the section below.

Response : correction done in the revised version of the manuscript.

- On page one they refer to the radiofrequency chirp $\alpha/2\pi$ without first explaining what the concept of the measurement is. I suppose, in the atom interferometry community everybody will now it. Then you need not even define it. For all others, shifting it a few lines down, would be helpful. The units (2π) are irrelevant in the text and sufficient in the graph.

Response : correction done in the revised version of the manuscript.

- The reader is given the millimeter extensions of the housing of the classical gravimeter (215 x 517 mm): I was surprised to see this, because it appears rather vain information, or do I overlook anything important here?

Of course, the overall size and weight of the entire “hybridized system” could be of interest (but even then decimeter accuracy suffices) to see how transportable it is.

Response : Centimeter extension are now reported in the revised version of the manuscript.

- The acceleration “a” should be written in italics, not to mix it with the indefinite article

Response : correction done in the revised version of the manuscript.

- Spelling and grammar still require strong revision in many places:

- Articles or plural endings are often missing - too often to list them all here. That may be quickly corrected.

Response : correction done in the revised version of the manuscript.

- attempts are usually “made” not “done”

Response : correction done in the revised version of the manuscript.

- in two instances the authors use the phrase we report ... to refer to work by other groups. This may be misleading, even though they properly cite the others. At least I would expect own work to follow such a phrase.

Response : correction done in the revised version of the manuscript.

- the continuous form “is corresponding” is not appropriate for something that is generally true or repeated etc. A native English reader will sort this out in a few minutes.

Response : correction done in the revised version of the manuscript.

- the citations are dominantly taken from the wide geodesy community. Since the paper does not report any new geodetic information but rather a technological advance in making an atom interferometer robust and isolated for a rough sea, one should probably add citations from the atom interferometry community, too. I was surprised too see almost only french groups cited, while US groups, German, Australian, British and more recently also Chinese groups have been pushing the field enormously. This should be taken into account.

Response : It is not the purpose of our paper focused on mobile gravimetry to have a state of the art in atom interferometry with the citation associated. The reference articles in atom

gravimetry have been cited and also the inertial measurements in a moving platform. We do not think that a more complete state of the art in atom interferometry is interesting for the readership of Nature communications.

Reviewers' comments:

Reviewer #1 (Remarks to the Author):

I agree that a gravity gradiometer is not a gravimeter, as the authors point out. It is in fact two gravimeters! This type of gradiometer gives both direct acceleration output as well as gradient output. Though the design constraints are somewhat different (stability of mirrors, e.g.) And though the truck moved slowly, this demonstration also occurred eight years prior. I imagine the specs of that device have now improved, not to mention sounding rocket tests and others that have happened more recently.

Moreover, this very journal has already published a mobile gravimeter!

"Detecting inertial effects with airborne matter-wave interferometry," R. Geiger, et al., Nat. Comm. 2, 474 (2011).

I think it does a disservice to the history of the field to ignore this prior work, as well as other references from groups that have developed AI gravimeter technology over the last few decades (Chu, Kasevich, Bouyer, Tino, Ertmer, etc.).

- > Our work reports for the first time mobile gravity measurement with
- > a matter wave sensor in a real operating environment. Moreover,
- > compared to a state of the art mobile gravimeter, improved precision
- > have been obtained. To our point of view, it is an important contribution here.

I think these results are a very nice technical achievement, but the number of qualifications needed to make such a "first" statement is troubling. One needs to parse "mobile" and "real operating environment" to exclude trucks (Stanford 2008), planes (AOSense 2010, Bouyer/Landragin 2011), and rockets (MAIUS-1 2016).

Again, I find this to be a solid work and advancement of gravimeter technology, but more suited to a technical journal in navigation or geodesy.

Reviewer #2 (Remarks to the Author):

Having re-read the authors comments it seems it all boils down to two aspects:

1. The perception of "importance"

This is a key in all papers and a matter of personal weighing of facts. It seems that both reviewers unanimously saw the scientific importance not yet demonstrated in full and voted against publication.

Is the technical feat of putting the atom gravimeter on a ship rather than a plane, a truck or step by step to different positions in the country substantially new? Technologically, this is a yes, since there is a lot of refinement in isolation needed to achieve this. Scientifically, this still holds a question mark. This does, in no way, reduce the technical merits - of course everybody would feel shaky on high amplitude waves and it is impressive that the measurements worked, even in such an environment. Because of that, and on second thought and even though the paper has hardly changed in content, I tend to reconsider my view on the interest for Nature Comm.

This still leaves numerous points open for discussion that I strongly recommend to reconsider:

The authors and referees seem not to entirely agree on the definition of 'mobile gravimeter'. To me it is the "first gravity measurement with a matter wave sensor in a mobile platform in a ship" but not a "first gravity measurement with a matter wave sensor in a mobile platform" as such. The slow truck (rev 1) is a counter example as is the muQuans system that is fully "mobile" but not usually moving when being run and therefore measuring microGal rather than mGal, cited as a standard here.

Interestingly, the examples cited in the authors' rebuttal, namely ice melting, volcano monitoring and ground water variation, are key applications of mobile but "not moving" gravimeters - such as the commercial system by MuQuans. The improvements required for operation on a ship are not needed there and the accuracy reached on the boat cannot compete with the (mobile but non-moving) systems.

2.

I am somewhat surprised by the attitude of the authors, when they respond to a request for clearer, more extensive information, that they themselves find all nice or useful. A paper should serve the community more than vanity and if the external feedback suggests that the desired information does not come across, why not try to improve?

I maintain that Fig. 3 and 5 are better suited for public outreach activities than for informing scientists. Having said that, one may also claim, that especially because of that the paper will be looked at by more than the specialists in the field.

I also maintain that it is less useful to specify the length of the boat (again this is more interesting for public outreach) than to give clear information about the amplitude, frequency and accelerations of the ship motion in all three directions.

Also: even if the term hybridized exists (which I never doubted), the suggestion only was that it appears to be over-emphasized for what it is. It would have been extremely easy to compromise here for the benefit of the reader without losing any quality.

Finally, it strikes me as almost unprofessional that the hint towards a more balanced citation strategy is not simply taken as granted and integrated (which would have cost nothing and would have emphasized their knowledge of the field) but simply rejected with the surprising insight that they think the reader of Nat Comm may not be interested.

To clear this up: I am a reader of Nat Comm. I am interested in being served the relevant information!

Reviewer #3 (Remarks to the Author):

Dear Authors,

I read your article with the great interest. I have the following remarks:

1- Page 1, right column: You claim absolute gravity is measured with the precision better than that of usual spring gravimeters. As spring gravimeters cannot sense absolute values of gravity, this sentence deserves some attention. Maybe spatial changes in gravity derived from observed absolute values outperform the accuracy of gravity changes measured by standard spring gravimeters used in marine gravimetry.

2- The relationship between the falling length and state of the observation environment is not quite clear: shorter lengths (and respective times) are more suitable for kinematic applications where large values of the observation noise is expected. Can you be more specific about it? Did you test other lengths?

3- Figure 2 actually contains three figures which results in some difficulties reading their descriptions. Maybe the font size could be increased somehow?

4- Page 2, right column: The performance of atom interferometry was estimated by analysing systematic effects of the sensor. FG5 absolute gravimeters seem to outperform atom interferometry by one order of magnitude. They are regularly tested during international comparison campaigns which offer an excellent opportunity for testing any kind of instrument sensing absolute gravity.

5- Figure 3 shows the test area in the Atlantic Ocean. Depth profiles along surveyed ship tracks would help to interpret gravity variations derived from observed values.

6- Figure 4 shows differences between forward and backward gravity measurements. It would be nice to see differences to "reference gravity" as well as their statistics and possibly also distributions. How would you explain two different mean values for forward and backward survey? Is there any drift present in the data of the cold atom gravimeter? The depth profile could also be plotted in this figure.

7- Figure 5 shows two anomalous gravity maps of the same area. One is derived from surveyed data, the other one represents gravity derived from satellite altimetry data. What is the spatial resolution of the satellite altimetry field? How would you estimate the effect of the GMT interpolation technique applied in this figure? It would be better to show the map of the differences and include their statistics as well.

8- Figure 6: Bad performance of satellite altimetry in coastal waters is well known. Why not to use any of the latest combined global gravity field models as a reference data instead?

9- It is remarkable what can be achieved in the field of gravity field mapping. To test kinematic applications of matter-wave sensors, I would prefer using ground vehicles as moving platforms since they can visit points with well defined absolute values of gravity.

Point-by-point response to the referees' comments

Reviewer #1 (Remarks to the Author):

I agree that a gravity gradiometer is not a gravimeter, as the authors point out. It is in fact two gravimeters! This type of gradiometer gives both direct acceleration output as well as gradient output. Though the design constraints are somewhat different (stability of mirrors, e.g.) And though the truck moved slowly, this demonstration also occurred eight years prior. I imagine the specs of that device have now improved, not to mention sounding rocket tests and others that have happened more recently.

Response

I understand remarks coming from reviewer 1 but I am sorry to insist that the gradiometer described in the thesis "*Mahadeswaraswamy, C. Atom interferometric gravity gradiometer: disturbance compensation and mobile gradiometry (2009)*" is not two gravimeters for two reasons. First, this gradiometer is measuring a horizontal component of the gravity gradient tensor da_x/dx and thus it is not sensitive to the vertical acceleration and thus can not measure the gravity acceleration. Second, if the gradiometer does not have a high performance vibration isolation platform or hybridization with classical accelerometers, there is no possibility to have direct acceleration outputs because vibration noise is blurring the interference fringes for each atom accelerometer. In the thesis, there is no direct acceleration measurement reported and no mention of high performance vibration isolation system or hybridization that would allow having a direct acceleration output.

It is true that the demonstration occurs 8 years ago, however there is nothing reported in the literature on an improved device.

Concerning the sounding rocket tests, which constitute in any case an impressive step in the field of atom interferometry, nothing has been published yet. The web page reports the demonstration of matter wave interference in space but no gravity measurements were achieved.

Moreover, this very journal has already published a mobile gravimeter!

"Detecting inertial effects with airborne matter-wave interferometry," R. Geiger, et al., Nat. Comm. 2, 474 (2011).

I think it does a disservice to the history of the field to ignore this prior work, as well as other references from groups that have developed AI gravimeter technology over the last few decades (Chu, Kasevich, Bouyer, Tino, Ertmer, etc.).

Response

The reference "Detecting inertial effects with airborne matter-wave interferometry," R. Geiger, et al., Nat. Comm. 2, 474 (2011)" does not report a mobile gravimeter. No gravity measurements are reported in this article. Only horizontal acceleration measurements are reported. We know very well this reference because we are coauthor.

In our manuscript, works of Chu, Kasevich and Bouyer on gravimeter technology have already been cited (ref 14, 17, 18, 19). In the revised version of the article, we add also references on Tino and Ertmer works.

I think these results are a very nice technical achievement, but the number of qualifications needed to make such a "first" statement is troubling. One needs to parse "mobile" and "real

operating environment" to exclude trucks (Stanford 2008), planes (AOSense 2010, Bouyer/Landragin 2011), and rockets (MAIUS-1 2016).

Response

Whatever the technology used, until now only relative sensors were available to measure gravity acceleration in a dynamic environment (ship, airplane). In our manuscript, it is the first time that a precise measurement of gravity acceleration is reported in dynamics with an absolute sensor. We think this is an important fact to mention in the manuscript. In the reference (Stanford 2008; Bouyer/Landragin 2011) or MAIUS-1 2016, no gravity acceleration measurements are reported. Concerning the reference AOSense 2010, we did not find this reference in the literature.

Reviewer #2 (Remarks to the Author):

The authors and referees seem not to entirely agree on the definition of 'mobile gravimeter'. To me it is the "first gravity measurement with a matter wave sensor in a mobile platform in a ship" but not a "first gravity measurement with a matter wave sensor in a mobile platform" as such. The slow truck (rev 1) is a counter example as is the muQuans system that is fully "mobile" but not usually moving when being run and therefore measuring microGal rather than mGal, cited as a standard here.

Response:

Yes, we agree the term "mobile" is confusing. In our manuscript, we never use this term to qualify our gravimeter.

We agree also that we report the "first gravity measurement with a matter wave sensor in a mobile platform in a ship" but not a "first gravity measurement with a matter wave sensor in a mobile platform" because of the slow truck. (ref. 28). We did not want to claim that in our manuscript but it seems that our message was not clear and can be misinterpreted because gravity can be measured by a gravimeter or a gradiometer and a moving vehicle can be interpreted by I move and then I measure in static and not by I measure while it is moving.

In the revised version of the manuscript, we try to be more cleared and to give a non confusing message.

In the introduction, we change "Here, we present the first absolute gravimeter able to measure precisely gravity from a moving vehicle." by "Here, we present the first absolute gravimeter able to measure gravity acceleration from a ship".

In the conclusion, we change

"In conclusion, we demonstrated for the first time ship borne gravity measurements with a matter wave sensor. This technology has allowed us to obtain for the first time an absolute gravity measurement in a moving vehicle and to improve the precision compared to a conventional spring gravimeter."

By

"In conclusion, we demonstrated sub-mGal ship borne gravity measurements with a matter wave sensor. This technology has allowed us to obtain for the first time absolute gravity measurements from a ship and to improve the precision compared to a conventional spring gravimeter."

Interestingly, the examples cited in the authors' rebuttal, namely ice melting, volcano monitoring and ground water variation, are key applications of mobile but "not moving" gravimeters - such as the commercial system by MuQuans. The improvements required for operation on a ship are not needed there and the accuracy reached on the boat cannot compete with the (mobile but non-moving) systems.

Response :

We agree that measuring gravity on a moving platform is two orders of magnitude less precise than measuring gravity in static. But mapping gravity with a static gravimeter is a very long operation and not all the places are accessible. Even if the precision is less good, mapping gravity on a moving platform like an airplane is very interesting because it is faster and measurements can be made in principle everywhere. The precision reached by airborne or marine gravimetry (~0.5 mGal) can address volcano or ice melting issue (see figure below and webpage: <http://www.ldeo.columbia.edu/res/pi/icebridge/Gravity.html> and <https://hal.archives-ouvertes.fr/hal-00535572/document>). For ground water variation study, the level of precision reached by airborne gravimetry is not yet sufficient. However, this is addressed by space gravimetry (Grace) and Space agency (ESA and NASA) are studying the use of a matter wave sensor for future space gravity mission. The work reported in our manuscript and particularly the hybridization with a classical sensor could help designing a space matter wave sensor.

Figure 1.2 Gravity accuracy and resolution requirements for geophysical applications: accuracy in mGal is denoted on the horizontal axis, with resolution on the vertical

Figure extracted from the thesis of Sandra L. Kennedy: "Acceleration Estimation from GPS Carrier Phases for Airborne Gravimetry" (2002).

I maintain that Fig. 3 and 5 are better suited for public outreach activities than for informing scientists. Having said that, one may also claim, that especially because of that the paper will be looked at by more than the specialists in the field.

Response

We agree that on Fig. 1 and 3, there is no scientific information for an atom interferometer specialist but Fig.1 could be interested for a gravimeter user because it gives an overview of the cold atom gravimeter and allows comparing its size with a spring gravimeter. Fig.3 could be interested for a geophysicist because it gives an overview of the location of the gravimetry survey and the distance relative to the coast which is important for the comparison with gravity satellite maps. However, these figures are not essential for our manuscript and if the editor thinks that they are unnecessary, we will remove them.

I also maintain that it is less useful to specify the length of the boat (again this is more interesting for public outreach) than to give clear information about the amplitude, frequency and accelerations of the ship motion in all three directions.

Response

We add in the text of the revised version of our manuscript information about the frequency and the amplitude of the acceleration in the boat. The amplitude of acceleration was already given in Figure 7 a) in the methods section.

Also: even if the term hybridized exists (which I never doubted), the suggestion only was that it appears to be over-emphasized for what it is. It would have been extremely easy to compromise here for the benefit of the reader without losing any quality.

Response

In the revised version of our manuscript, we replace the term hybridization by combination.

Finally, it strikes me as almost unprofessional that the hint towards a more balanced citation strategy is not simply taken as granted and integrated (which would have cost nothing and would have emphasized their knowledge of the field) but simply rejected with the surprising insight that they think the reader of Nat Comm may not be interested.

To clear this up: I am a reader of Nat Comm. I am interested in being served the relevant information!

Response

We add in the revised version of our manuscript some words about atom interferometry with associated citations of the main groups working on this subject.

Reviewer #3 (Remarks to the Author):

1- Page 1, right column: You claim absolute gravity is measured with the precision better than that of usual spring gravimeters. As spring gravimeters cannot sense absolute values of gravity, this sentence deserves some attention. Maybe spatial changes in gravity derived from observed absolute values outperform the accuracy of gravity changes measured by standard spring gravimeters used in marine gravimetry.

Response

Yes, we agree spring gravimeter cannot sense absolute values of gravity. In the revised version of our manuscript, we change usual spring gravimeter by usual calibrated spring gravimeter. For our survey the spring gravimeter was calibrated with a reference gravity point at the harbor before and after the campaign and thus one can deduce "absolute gravity" from the spring gravimeter measurements.

2- The relationship between the falling length and state of the observation environment is not quite clear: shorter lengths (and respective times) are more suitable for kinematic applications where large values of the observation noise is expected. Can you be more specific about it?

Did you test other lengths?

Response

In static condition, longer is the atom interrogation time, better will be the precision of the gravity measurement. This is because the scale factor is proportional to square of the interrogation time ($\phi = k_{\text{eff}} a T^2$). In dynamic condition, we can not use the biggest interrogation time possible with our sensor because of the limitation of the hybridization between the force balanced accelerometer and the atom accelerometer. Indeed the hybridization is working well only if the error on the estimated acceleration by the force balanced accelerometer is smaller than the period of output of the atom accelerometer $\lambda/2T^2$ (see method section). In dynamic environment, when the sensor is subject to large variation of acceleration, uncertainty of the transfer function of the force balanced accelerometer leads to an important error to the estimated acceleration given by the force balanced accelerometer. Thus in dynamic environment, the interrogation time T is decreased until the error of the estimated acceleration is smaller the period of the atom accelerometer: $\lambda/2T^2$. That is why during our marine campaign, the interrogation time used was 10 ms or 20 ms and only the shorter falling distance was used while the sensor could use an interrogation time up to 39 ms.

3- Figure 2 actually contains three figures which results in some difficulties reading their descriptions. Maybe the font size could be increased somehow?

Response

The font size has been increased in the revised version of the manuscript

4- Page 2, right column: The performance of atom interferometry was estimated by analysing systematic effects of the sensor. FG5 absolute gravimeters seem to outperform atom interferometry by one order of magnitude. They are regularly tested during international comparison campaigns which offer an excellent opportunity for testing any kind of instrument sensing absolute gravity.

Response

Yes, FG5 outperform our atom gravimeter by one order of magnitude. However dedicated atom gravimeters for static condition have similar performance than FG5. (see ref 15 for a comparison with a FG5). Compared to static atom gravimeter, our onboard atom gravimeter has a smaller falling distance (~ 10 times) that is why the performance in static does not reach the FG5 performance. However our sensor is well adapted for dynamic measurement where the precision goal is only 0.5 mGal.

5- Figure 3 shows the test area in the Atlantic Ocean. Depth profiles along surveyed ship tracks would help to interpret gravity variations derived from observed values.

Response

In the revised version of the manuscript, we add Depth profiles along surveyed ship tracks.

6- Figure 4 shows differences between forward and backward gravity measurements. It would be nice to see differences to "reference gravity" as well as their statistics and possibly also distributions. How would you explain two different mean values for forward and backward survey? Is there any drift present in the data of the cold atom gravimeter? The depth profile could also be plotted in this figure.

Response

In the revised version of the article, we add the "reference gravity" on figure 4 and the differences between the measured and the reference gravity. The statistics on the difference are given in Table 1. The two different mean values for forward and backward survey could be explained by the tide effect. Indeed we can see on figure 5 of the revised manuscript, that the difference of mean value comes essentially from the shallow part (40-80 km) where tides are bigger.

We did not notice drift of the atom gravimeter. The gravity measurements in Brest harbor in October 2015 and January 2016 are in agreement within 0.3 mGal. This small difference can be attributed to the fact that the ship was not exactly at the same place.

In the revised version of the article, the depth profile is plotted in fig. 4.

7- Figure 5 shows two anomalous gravity maps of the same area. One is derived from surveyed data, the other one represents gravity derived from satellite altimetry data. What is the spatial resolution of the satellite altimetry field? How would you estimate the effect of the GMT interpolation technique applied in this figure? It would be better to show the map of the differences and include their statistics as well.

Response

The spatial resolution of the satellite altimetry data is 16 km.

The effect of the interpolation technique can be analyzed by looking at the difference between the ship borne model which interpolates the gravity measurements and the satellite model (see Figure 5 c) on the revised version of the manuscript). Expect the left and the right border, the differences along the measurement lines and outside the measurement lines are similar. Thus the effect of the GMT interpolation technique seems weak in this area. However, on the left and right border, one can notice an important difference this is probably due to the effect of the interpolation technique.

The statistics (mean and standard deviation) of the difference between the satellite model and the ship borne gravity measurement are given in the text.

8- Figure 6: Bad performance of satellite altimetry in coastal waters is well known. Why not to use any of the latest combined global gravity field models as a reference data instead?

Response

We did the comparison with the combined model eigen6C4. Although it is better in coastal area, it is not satisfactory compared to ship borne measurements.

As Sandwell, EIGEN is also limited and not very precise in coastal area especially in small area like Douarnenez bay.

The comparisons between Sandwell and Eigen6C4 on the Meriadzec area and along the calibration profile show the benefit of Sandwell compared to EIGEN6C4. That is why we choose Sandwell for our study.

9- It is remarkable what can be achieved in the field of gravity field mapping. To test kinematic applications of matter-wave sensors, I would prefer using ground vehicles as moving platforms since they can visit points with well defined absolute values of gravity.

Response

Yes, testing matter wave sensor in ground vehicles like a truck is very interesting because of the possible comparison with defined absolute values of gravity. However, our gravimeter has been designed for marine environment i.e. slow variation of acceleration and attitude (~ 0.15 Hz) which is very different from a truck environment. In order to obtain precise measurements in a truck, our gravimeter may need some adaptations.

REVIEWERS' COMMENTS:

Reviewer #1 (Remarks to the Author):

I am afraid I just cannot come around to seeing the new scientific achievements in the way the authors do. Again, this is a very nice technical work, but I do not find it to be a significantly new enough achievement to meet the standards of Nature Comm, as I see them.

The fact the novelty of the results requires so much parsing of language I think is a sign that this is technical achievement for the applied inertial measurement community, but not a novel achievement for a wider scientific community.

Reviewer #2 (Remarks to the Author):

The authors have responded favorably to the majority of comments and requests. I think the paper will find an interested audience both in the general public and among atom interferometer specialists. "go ahead" from my side.

Reviewer #3 (Remarks to the Author):

I read all your rebuttals with great interest. Thank you for considering my suggestions which may have improved the text.

Point-by-point response to the referees' comments

REVIEWERS' COMMENTS:

Reviewer #1 (Remarks to the Author):

I am afraid I just cannot come around to seeing the new scientific achievements in the way the authors do. Again, this is a very nice technical work, but I do not find it to be a significantly new enough achievement to meet the standards of Nature Comm, as I see them.

The fact the novelty of the results requires so much parsing of language I think is a sign that this is technical achievement for the applied inertial measurement community, but not a novel achievement for a wider scientific community.

Response

We thank the reviewer for the careful reading of our manuscript and the valuable suggestions which improved our article.

However, we have still a different point of view of the impact of our paper for a wide scientific community.

Reviewer #2 (Remarks to the Author):

The authors have responded favorably to the majority of comments and requests. I think the paper will find an interested audience both in the general public and among atom interferometer specialists.
"go ahead" from my side.

Response

We thank the reviewer for the careful reading of our manuscript and the valuable suggestions which improved our article.

.

Reviewer #3 (Remarks to the Author):

I read all your rebuttals with great interest. Thank you for considering my suggestions which may have improved the text.

Response

We thank the reviewer for the careful reading of our manuscript and the valuable suggestions which improved our article.